# Integrative Analysis Identifies TCIRG1 as a Potential Prognostic and Immunotherapy-Relevant Biomarker Associated with Malignant Cell Migration in Clear Cell Renal Cell Carcinoma

**DOI:** 10.3390/cancers14194583

**Published:** 2022-09-21

**Authors:** Chao Xu, Bolin Jia, Zhan Yang, Zhenwei Han, Zhu Wang, Wuyao Liu, Yilong Cao, Yao Chen, Junfei Gu, Yong Zhang

**Affiliations:** 1Department of Urology, The Second Hospital of Hebei Medical University, 215 Heping West Road, Shijiazhuang 050000, China; 2National Cancer Center, National Clinical Research Center for Cancer, Hebei Cancer Hospital, Chinese Academy of Medical Sciences, Jinyuan Road, Economic, and Technological Development Zone, Guangyang District, Langfang 065001, China; 3Molecular Biology Laboratory, Talent and Academic Exchange Center, The Second Hospital of Hebei Medical University, 215 Heping West Road, Shijiazhuang 050000, China; 4National Cancer Center, National Clinical Research Center for Cancer, Cancer Hospital, Chinese Academy of Medical Sciences, Peking Union Medical College, 17 Panjiayuan Nanli, Chaoyang District, Beijing 100021, China

**Keywords:** kidney renal clear cell carcinoma, treatment, immune infiltration, prognostic biomarker, TCIRG1, migration

## Abstract

**Simple Summary:**

TCIRG1, also known as V-ATPase-a3, is critical for cellular metabolism, membrane transport, and intracellular signaling through its dependent acidification. In earlier research, TCIRG1 was found to be dysregulated in several cancers and to accelerate the growth of various malignancies. The molecular mechanisms behind TCIRG1 and its possible role in the development of clear cell renal cell carcinoma are still poorly understood. Our research is the first to thoroughly examine TCIRG1’s function in clear cell renal cell carcinoma prognosis, immunity, and treatment. The validity that TCIRG1 can accelerate the development of renal clear cell carcinoma was also confirmed in this work by using certain testable experiments. This establishes the theoretical framework for our future investigation into the occurrence and progression of clear cell renal cell carcinoma.

**Abstract:**

**Background:** TCIRG1, also known as V-ATPase-a3, is critical for cellular life activities through its dependent acidification. Prior to the present research, its relationship with prognostic and tumor immunity in clear cell renal cell carcinoma (ccRCC) had not yet been investigated. **Methods:** We assessed TCIRG1 expression in normal and tumor tissues using data from TCGA, GEO, GTEX, and IHC. We also analyzed the relationship between TCIRG1 and somatic mutations, TMB, DNA methylation, cancer stemness, and immune infiltration. We evaluated the relevance of TCIRG1 to immunotherapy and potential drugs. Finally, we explored the effect of TCIRG1 knockdown on tumor cells. **Results:** TCIRG1 was overexpressed in tumor tissue and predicted a significantly unfavorable clinical outcome. High TCIRG1 expression may be associated with fewer PBRM1 and more BAP1 mutations and may reduce DNA methylation, thus leading to a poor prognosis. TCIRG1 was strongly associated with CD8+ T-cell, Treg, and CD4+ T-cell infiltration. Moreover, TCIRG1 was positively correlated with TIDE scores and many drug sensitivities. Finally, experiments showed that the knockdown of TCIRG1 inhibited the migration of ccRCC cells. **Conclusions:** TCIRG1 may have great potential in identifying prognostic and immunomodulatory mechanisms in tumor patients and may provide a new therapeutic strategy for ccRCC.

## 1. Introduction

Clear cell renal cell carcinoma (ccRCC) is typical primary malignancy. It is second only to bladder cancer in terms of mortality from urological cancers, with an insidious onset and resistance to radiotherapy and chemotherapy [1,2]. ccRCC is the most prevalent form of RCC, accounting for between 70% and 80% of cases. Tyrosine kinase inhibitors (TKIs), immune checkpoint inhibitors (ICIs), and molecule-targeting medications have been used more frequently in advanced ccRCC [3]. Furthermore, in advanced ccRCC, a highly dynamic, adaptable, and heterogeneous tumor microenvironment (TME) may induce drug resistance [4]. Therefore, biomarkers associated with an immunosuppressive TME in patients with advanced ccRCC should be urgently investigated. V-ATPases, which are complexes of several subunits that act as ATP-driven proton pumps, are present in the inner membranes of mammalian cellular compartments and the plasma membranes of several specialized cells; these are involved in various critical biological processes [5]. These processes depend on V-ATPase control of the acidity gradient in the Golgi apparatus for glycosyltransferases needed for the activities to occur [6]. In addition, the pH of endosomes, lysosomes, and secretory vesicles also depends on regulation by V-ATPases [7]. The a, c, c’, d, and e subunits are present in the proton translocation domain of V-ATPases, whereas the AH subunit is present in the ATP hydrolysis domain of V1 [6]. The subcellular localization of V-ATPase is controlled by the V-ATPase subunit “a”, which has four isoforms (a1–a4). The a3 subunit is also referred to as T-cell immunoregulatory factor 1 (TCIRG1) [8].

Evidence from numerous studies has indicated that TCIRG1 promotes the development of numerous malignancies [9,10,11,12]. For example, in a study by Yang et al. [12], the aberrant overexpression of TCIRG1 in patients with recurrent HCC undergoing total hepatectomy and knockdown of TCIRG1 inhibited tumor cell growth and proliferation in HCC cell lines, induced cell death, and promoted tumor migration via the EMT. McGuire et al. [9] showed the knockdown of TCIRG1, which decreased lung and bone metastases, MMP-2 and MMP-9 expression in B16-F10 cells, and invasiveness and migration. However, the potential impact of TCIRG1 in the development of RCC and the underlying molecular mechanisms remain unknown.

In this study, we combined data from TCGA and GEO public databases and IHC experiments to validate the expression of TCIRG1 in ccRCC samples and its clinical significance. The molecular changes and immunological characteristics of TCIRG1 expression were also investigated, and its effects on clinical outcomes were evaluated using various bioinformatics techniques. The findings suggested that TCIRG1 may serve as a predictive biomarker for ccRCC and an immunological target for the selection of patients with ccRCC who respond to ICI in the future.

## 2. Materials and Methods

### 2.1. Analysis of the Connection between Prognosis and TCIRG1

In this work, 356 renal somatic mutation profiles, 535 renal samples, 72 normal renal samples, and related clinical data from the TCGA (https://portal.gdc.cancer.gov/, accessed on 22 February 2022) RNA-sequencing project were retrieved using UCSC Xena (https://xenabrowser.net/, accessed on 22 February 2022). For conducting the differential analysis of tumor and normal tissues, we used data obtained from normal tissue samples from TCGA and GTEX (http://commonfund.nih.gov/GTEx/, accessed on 22 February 2022) and tumor cell data from TCGA. Additionally, we investigated TCIRG1 expression in RCC using UALCAN (http://ualcan.path.uab.edu/, accessed on 25 February 2022). To verify the differential expression of TCIRG1 in normal tissues and KIRC (kidney renal clear cell carcinoma) tissues, we used GSE66272 [13,14], GSE53757 [15], and GSE15641 [16]) as the validation set. The gene expression profiles from GSE66272, GSE53757, and GSE15641 were obtained from the GEO (https://www.ncbi.nlm.nih.gov/geo/, accessed on 12 March 2022) database. Following this, we used ggplot2 (v3.3.2) and R software version 4.0.3 to analyze TCGA clinical data. *p*-values less than 0.05 were considered statistically significant. We assessed the clinical predictive value of TCIRG1 expression in multiple cancers using the “survival” package. The diagnostic utility of TCIRG1 expression in the RCC sample was assessed using receiver operating characteristic (ROC) curves. The higher the value of the area under the ROC curve (AUC), the higher the corresponding predictive power. Finally, we used univariate and multivariate cox regression analyses and the “forestplot” package to display each variable *p*-value, Hazard Ratio (HR), and 95% Confidence Interval (CI) in forest plots. The overall recurrence rate for the following 1, 3, and 5 years was predicted using a nomogram with the “rms” package based on the findings of multivariate Cox proportional risk analysis. The nomogram shows the results of these factors graphically and enables the computation of the prognosis risk for each patient by using the points assigned to each risk factor.

### 2.2. Construction of the TCIRG1 Co-Expression Network and Annotation of Its Associated Genes

Using TCGA, RNA sequencing information for KIRC samples was retrieved. The “limma” R program was used to compute co-expressed genes, and Pearson’s correlation was used to verify relationships. False Discovery Rate (FDR) < 0.01 and Pearson’s correlation (absolute value) > 0.4 were used to construct the co-expressed gene network. Then, we used the R package “ClusterProfiler” for the Gene Ontology biological process (GO_BP), Cellular component (GO_CC) and Molecular function (GO_MF) of TCIRG1 and its co-expressed genes, and the Kyoto Encyclopedia of Genes and Genomes (KEGG).

### 2.3. Correlation of TCIRG1 with Molecular and Immunological Properties

The total amount of somatic gene coding mistakes, base substitutions, gene insertion, or deletion errors that take place inside a certain genomic region is known as the “tumor mutation load” (TMB). From the TCGA dataset, we were able to extract the relevant somatic alteration data for the TCGA-KIRC cohort. The number of somatic nonsynonymous point mutations in each sample was then calculated using the “maftools” package in R software. The association between TCIRG1 and TMB was described using Spearman’s correlation analysis. One of the most frequent and well-researched epigenetic processes, DNA methylation, is essential for transcriptome changes and the dysregulation of cellular pathways in malignancies [17]. To examine TCIRG1 DNA methylation in TCGA-KIRC, we analyzed the relationship between DNA methylation and TCIRG1 expression. Survival analysis based on TCIRG1 DNA methylation levels was studied using KM survival curves. In addition to being crucial for tumor growth, cancer stemness may also affect immune cells in the milieu around the tumor [18]; thus, RNA sequencing based on mRNA expression and DNA sequencing based on DNA methylation were used to determine the stemness of the tumor [19]. In addition, we analyzed TCGA gene expression data by using the “CIBERSORT” package to assess the abundance of 22 different immune cell subpopulations [20]. The “immunedeconv” package was used to evaluate the quantity of 10 different immune cell subpopulations [21]. We also used the Expression Data (ESTIMATE) algorithm to evaluate the Immunescores, Stromalscores, and Estimatescores [22]. TCIRG1 and immune-related genes (IRGs) co-expression analysis was carried out by using GEPIA2 (http://gepia2.cancer-pku.cn/#index, accessed on 18 March 2022) [23]. Additionally, TCIRG1 expression and other immune cell surface markers were investigated by using Timer2.0 (http://timer.cistrome.org/, accessed on 18 March 2022) [24].

### 2.4. Exploration of Immunotherapy for ccRCCs

To predict immunological escape and immunotherapy receptivity in the groups with high and low TCIRG1 expression, we used the Tumor Immune Dysfunction and Exclusion (TIDE) website (http://tide.dfci.harvard.edu, accessed on 20 March 2022) to predict the responsiveness of patients with data in the TCGA-KIRC cohort to immunotherapy. Subsequently, we obtained data from GEO for GSE67501 (Expression data in human renal cell carcinoma samples from patients who did or did not respond to anti-PD-1 immunotherapy) to analyze the responsiveness of TCIRG1 to immunotherapy. In addition, using the “pRRophetic” package, the half-maximal inhibitory doses (IC50) of medications with varying sensitivities across groups with high and low TCIRG1 expression were determined [25].

### 2.5. siRNAs Transfection

TCIRG1 expression was reduced using siRNA-TCIRG1#1 (GenePharma) and siRNA-TCIRG1#2 (GenePharma), non-specific siRNA-NC (GenePharma) was used as a control, and Lipofectamine RNAiMAX was used according to the manufacturer’s instructions. 769P cell lines and Caki-1 cell lines were incubated with siRNA complexes for 48 h before proceeding to the next step. The involved siRNAs’ sequences are listed below:

siRNA-TCIRG1#1 sense:5′-GGACCUGAGGGUCAACUUUTT-3′

antisense:5′-AAAGUUGACCCUCAGGUCCTT-3′;

siRNA-TCIRG1#2 sense5′-CCAUCUACACCGGCUUCAUTT-3′

antisense5′-AUGAAGCCGGUGUAGAUGGTT-3′;

siRNA-NC sense5′-GGGUGGAAUUCCAGAACAATT-3′

antisense5′-UUGUUCUGGAAUUCCACCCTT-3′.

### 2.6. Detection of mRNA Expression by Quantitative Fluorescence Polymerase Chain Reaction (qRT-PCR)

RNAs were isolated using the Trizol reagent (Invitrogen, Grand Island, NY, USA). RNA was utilized for reverse transcription using Superscript III transcriptase (Invitrogen). In order to measure the mRNA expression level of the target genes, quantitative real-time PCR (qRT-PCR) was carried out using a Bio-Rad CFX96 system with SYBR green. Expression levels were adjusted to account for the expression of GAPDH. The qRT-PCR procedure was as follows: 50 °C for two minutes, 95 °C for eight minutes and thirty seconds, 45 cycles at 95 °C for fifteen seconds each, and 60 °C for one minute; 95 °C for 1 min, 55 °C for 1 min, and 55 °C for 10 s made up the extension. The normalized control was GAPDH. The involved primer sequences are listed below:

TCIRG1 5′-CCCCATCTTTGCCGCCTTT-3′

TCIRG1 3′-CCCGTGCCTGAGTAGAACT-5′;

GAPDH 5′-TGTGGGCATCAATGGATTTGG-3′

GAPDH 3′-ACACCATGTATTCCGGGTCAAT-5′.

### 2.7. Western Blotting Assay

RIPA whole-cell lysis solution was used to lyse the cells. Total proteins were then extracted, measured, and semi-dry transferred to PVDF (Millipore, Billerica, MA, USA) membranes after separation by 8–12% SDS-PAGE electrophoresis; PVDF membranes were first closed with TBS + Tween (TBST) solution containing 5% skim milk powder for 2 h, washed, and incubated with primary antibody at 4 °C overnight (TCIRG1, 12649-1-AP, Proteintech, 1:50); they were then rewashed and incubated for 2 h with a secondary antibody that was colored by horseradish peroxidase. The PVDF membrane was washed before the chemiluminescent substrate was applied and then developed by a gel imaging system, after which the grayscale values were measured.

### 2.8. Transwell Migration Experiments

The cells were collected with serum-free media and plated into the upper chambers of 8.0 µm pore size polycarbonate membrane filters (CorningIncorporated, Corning, NY, USA) at 1 × 105/mL, and 600 μL 10% FBS media was added into the lower chambers for incubation at 37 °C for 36 h. Three replicate wells were set up for each group of experiments, and the same experiments were repeated three times. The field of view was randomly selected for counting, and pictures were taken at the same time. The number of migrating cells was measured with Image J software.

### 2.9. Wound-Healing Assay to Detect Cell Migration

The 6-well plate’s back was marked with three parallel lines before the tumor cells were added at a density of 1 × 10^5^ cells per well and transfected with si-TCIRG1 and si-NC, respectively, after the cells had adhered entirely to the wall. After 24 h, the growth of tumor cells was observed, and the tumor cells were scratched with a 10 uL gun in the 6-well plate perpendicular to the horizontal line on the back, washed with phosphate buffer solution (PBS), and added to a serum-free medium at 37 °C in a 5% CO_2_ incubator. The culture medium was placed in a 37 °C, 5% CO_2_ incubator. The width of the scratch was measured by Image J software, and the cell migration rate was calculated.

### 2.10. Immunohistochemical Stainings and Evaluation

Briefly, 4 μm paraffin-embedded tissue cross sections were subjected to immunohistochemical staining. Sections were deparaffinized with xylene, rehydrated, pre-incubated with 10% normal goat serum (710027, KPL, USA), and then incubated overnight at 4 °C with primary antibody (TCIRG1, 12649-1-AP, Proteintech, 1:50) and washed. This was followed by treatment with secondary antibodies (horseradish peroxidase-labeled rabbit IgG antibody, 021516, KPL, USA). The DAB substrate kit was used for the color development reaction. Finally, sections were re-stained with hematoxylin and then dehydrated, cleared, and evaluated.

### 2.11. Statistical Analysis

R (version 4.03) and GraphPad Prism 8.0 software were used for statistical analysis and graphical visualization of the data. Correlations between variables were calculated using Pearson or Spearman coefficients. For all statistical calculations, *p* values less than 0.05 were regarded as statistically significant.

## 3. Results

### 3.1. Pan-Cancer Analysis of TCIRG1 Expression

We initially assessed TCIRG1 mRNA expression in 28 human cancer and normal tissues using a combination of data from the TCGA and GTEX datasets to investigate the potential involvement of TCIRG1 in carcinogenesis and progression. The pan-cancer examination of TCIRG1 expression revealed that the expression of TCIRG1 was higher in some cancer tissues than in healthy tissues, including BLCA, CHOL, GBM, HNSC, KIRC, and LGG. Additionally, TCIRG1 expression was downregulated in 20 tumor tissues, including ACC, BRAC, CESC, and COAD (Figure 1A). Next, we validated the TCGA dataset with paraneoplastic tissue samples > 30 in cancer types for paired difference analysis, the results of which are shown in Figure 1B. Subsequently, we compared the expression of TCIRG1 in tumor and normal samples using UALCAN. TCIRG1 expression was elevated in KIRC (Figure 1C). In addition, we used microarray data retrieved from the GEO database further to validate the expression of TCIRG1 in KIRC samples. TCIRG1 expression was elevated in the GSE66272, GSE53757, and GSE15641 samples, which agrees with the findings from the analysis of the TCGA cohort (Figure 1D–F). To further explore TCIRG1 expression in RCC, we used IHC to detect TCIRG1 expression at the protein level. IHC staining for TCIRG1 was negative in normal kidney specimens and positive in KIRC specimens (Figure 1H); the statistical chart of the results of IHC is shown in Figure 1G. Thus, we hypothesized that TCIRG1 is an unfavorable prognostic biomarker in renal cancer.

### 3.2. Poor Prognosis for KIRC Is Indicated by High TCIRG1 Expression

Since TCIRG1 expression differs significantly in various tumors and normal tissues, we believe that elevated TCIRG1 expression is significantly associated with a low OS, DSI, and PFI in KIRC, as indicated by the TCIRG1 mRNA expression patterns (Figure 2A,C,E). The corresponding risk values are shown in forest plots (Figure 2B,D,F). In addition, 33 human tumors showed significant (*p* < 0.05) OS, PFI, and DSS (Appendix A). Collectively, these results suggest that TCIRG1 is a risk factor for predicting poor prognosis in KIRC.

### 3.3. Correlation between High TCIRG1 Expression and Clinical Characteristics

To further elucidate the role of TCIRG1 expression in the progression of ccRCC, we used data from the TCGA cohort and investigated the correlation of clinical parameters with TCIRG1 expression. TCIRG1 expression varied drastically depending on the clinical stage of the tumor (Figure 3A–E). The stage, grade, T, M, and N were significantly different. A progression in the clinical stage (T, N, and M clinical stages) was correlated with an increase in TCIRG1 expression (all *p* < 0.05). A clinical correlation heat map can better present the results listed above (Figure 3F). The diagnostic potential of TCIRG1 for KIRC was also estimated using the ROC curve (Figure 3G), and an AUC of 0.919 (*p* < 0.05) was observed. In addition, to assess the independent prognostic predictors associated with the OS, both univariate and multivariate Cox regression were used. High TCIRG1 expression, as determined in the univariate study, grade, stage, and age significantly predicted poor OS. In addition, multivariate analysis revealed that age, grade, stage, and TCIRG1 expression were independent prognostic indicators for an unfavorable OS in KIRC (Table 1). To further highlight the prognostic potential of TCIRG1, a nomogram was created using the findings of multivariate Cox regression (Figure 3H). As shown in (Figure 3I), the calibration curves show that the 1-year, 3-year, and 5-year OS predictions and the actual data have good agreement. In conclusion, our results suggest that TCIRG1 may be an important prognostic indicator for predicting OS in patients with KIRC.

### 3.4. Molecular Characterization of Different TCIRG1 Subgroups

To better understand genetic changes and epigenetic modifications, we identified the top 10 genes with the greatest mutation rates in KIRC samples across high- and low-TCIRG1 expression subgroups. We then represented the data with waterfall plots (Figure 4A,B). In addition, we further investigated the differences in TCIRG1 expression between wild-type and mutant subgroups and found that higher TCIRG1 expression may result in fewer PBRM1 mutations and more BAP1 mutations (Figure 4C,D). The link between TCIRG1 expression TMB and DNA methylation was subsequently investigated; the outcomes are shown in Figure 4E,F. No correlation was observed between TCIRG1 expression and TMB (Cor = −0.019, *p* = 0.73), and a negative correlation was observed with DNA methylation (Cor = −0.23, *p* < 0.001). Survival analysis for assessing high- and low-TCIRG1 DNA methylation subgroups also showed that the high-DNA methylation subgroup had a better prognosis (Figure 4G). To predict tumor stemness, we employed RNAss based on mRNA expression and DNAss based on DNA methylation (Figure 4H,I). Notably, linear regression analysis showed no correlation between the DNAss levels and TCIRG1 expression (Cor = −0.11, *p* < 0.054), whereas the RNAss levels were negatively correlated with TCIRG1 expression (Cor = −0.27, *p* < 0.001). However, the TCIRG1 RNAss levels were not statistically significant as an indicator of OS (Figure 4J).

### 3.5. TCIRG1 Co-Expression Networks in KIRC

We analyzed the co-expression network of TCIRG1 in the TCGA-KIRC cohort. As shown in Figure 5A, 2093 genes were significantly associated with TCIRG1 expression (FDR < 0.01, COR > 0.4), of which 415 (green dots) were negatively associated genes and 1678 were positively associated genes (red dots). Figure 5 shows the heatmap with the fifty genes with the greatest correlation in expression. GO annotation revealed that these genes are involved in various immune responses, including T cell activation, leukocyte cell–cell adhesion, negative regulation of immune response, and regulation of mononuclear cell proliferation, among other functions (Figure 5C). Following this, we conducted KEGG pathway analysis and showed that the co-expression of genes in HIV-1 infection, the mTOR signaling pathway, the *PDL1* expression pathway, *PDL1* expression, and the *PD-1* checkpoint pathway in cancer and T cell receptor signaling pathway were enriched (Figure 5D). These findings imply that by controlling immunological responses, TCIRG1 expression may play a critical role in the development of human cancer.

### 3.6. Correlation of TCIRG1 Expression Levels with TME Immunity and Estimate Scores

The interaction between the TME and tumor has important implications in tumor development and the efficacy of immunotherapy. Based on transcriptomic data from TCGA, we used ESTIMATE to calculate the Immunescores, Stromalscores, and Estimatedscore of ccRCC tissues. Figure 6A demonstrates a correlation between the expression of TCIRG1 and the Immunescore and Estimatedscores of ccRCC. In addition, we found that higher Immunescores and Estimatescores were both associated with poor OS in patients with ccRCC (Figure 6B,C). These findings imply that the independent prognostic function of TCIRG1 in ccRCC may be related to changes in the TME.

### 3.7. Immune-Related Analysis of TCIRG1 in ccRCC

Using CIBERSORT and quanTIseq, the fraction of tumor-infiltrating immune cells (TIICs) was segmented in the TCGA cohort to further examine the link between TCIRG1 and immune cells in ccRCC tissue. Pearson’s correlation analysis revealed that the top three largest fractions of immune cells in CIBERSORT were CD4+ memory resting T cells (19%), M2 macrophages (18%), and CD8+ T cells (18%) (Figure 7A). In addition, as shown in Figure 7E, TCIRG1 expression showed the most significant correlation with the regulatory T cell (Treg) population and CD4+ memory T cell population (r = 0.35, −0.36, *p* < 0.001). In quanTIseq, neutrophils accounted for approximately 25%, M2 macrophages for approximately 14%, cells for approximately 13%, and CD8+ T cells for approximately 11% of the population (Figure 7B). TCIRG1 expression showed the most significant correlation with the Treg, neutrophil, NK cell, and CD4+ memory T cell populations (r = 0.47, 0.43, −0.44, −0.39, *p* < 0.001) (Figure 7F). Using the CIBERSORT and quanTIseq algorithms, we also investigated the link between TCIRG1 expression and immune cell infiltration as well as between immune cell populations in KIRC. The results are displayed in the correlation heat map (Figure 7C,D). Lastly, we used GEPIA2 [23] and TIMER2.0 [24] to assess the association between cell surface markers of different types of TIIC (Table 2); the results are consistent with the above results. Appendix A shows the relationship between TCIRG1 expression and TIICs in several other algorithms. The above results suggest that high TCIRG1 expression in ccRCC may be associated with increased renal Treg infiltration and CD4+ memory T cell depletion, for which the underlying mechanism remains unknown.

### 3.8. The Potential Role of TCIRG1 Expression in ccRCC Immunotherapy

We initially selected 28 genes as immune checkpoint signature genes. This helped us investigate the treatment strategies for KIRC. When compared to that in the low-TCIRG1 expression subgroup, the genes were upregulated in the high-TCIRG1 expression subgroup (Figure 8A). Next, using the TIDE website, we assessed how differently patients at various risk levels responded to immunotherapies. We found that samples in the high-expression subgroup had a very high potential for immune escape because the TIDE scores were significantly higher in the high-TCIRG1 expression subgroup than in the low-expression subgroup (Figure 8B). In addition, as shown in Figure 8C, TCIRG1 expression was significantly higher in the response group than in the non-response group in the RCC anti-*PD-1* immunotherapy dataset GSE67501 [26]. Finally, we screened potential drugs for ccRCC by predicting the IC_50_ of the drugs (Figure 8D–M). Individuals with data in the high-TCIRG1 expression subgroup had lower IC_50_ values for pazopanib, sorafenib, AKT inhibitor VIII, BMS-509744, JW-7-52-1, and thapsigargin, indicating that these patients were non-sensitive to these medications. In contrast, patients with data in the high-TCIRG1 expression category were more responsive to sunitinib, temsirolimus, 5-fluorouracil, and AP-24534.

### 3.9. The Potential of KIRC Cells to Migrate Is Inhibited in Response to TCIRG1 Knockdown

To investigate the role of TCIRG1 in KIRC, first, we selected the cell lines 769p and Caki-1 by western blotting tests to construct si-TCIRG1 cell lines (Figure 9A). Following this, we measured the protein levels of TCIRG1 using RT-PCR and protein blotting (Figure 9B–D) and performed wound-healing and transwell assays to investigate the effect of TCIRG1 expression on cell migration and invasion in KIRC (Figure 9E,F,I,J). The cell migration potential was significantly reduced in 769P and Caki-1 cell lines with TCIRG1 knockdown (Figure 9G,H,K,L). In conclusion, TCIRG1 may accelerate the development of KIRC by enhancing the migratory potential of cells (Figure 9A,D; uncropped original western blots can be viewed in Appendix A).

## 4. Discussion

Immune checkpoint inhibitors (ICIs) and anti-angiogenic agents have overturned the clinical treatment of tumors. Although antiangiogenic-drugs and ICIs have shown considerable efficacy in RCC, their role in advanced renal cell carcinoma remains limited [27]. The identification of biomarkers for predicting the response to ICI therapy and prognosis for patients with RCC is urgently needed; this problem has recently garnered significant interest in the scientific community.

*ATPases* are ATP-driven proton pumps are responsible for the acidification and maintenance of the pH of intracellular organelles. V-ATPase-dependent acidification is essential in various biological processes [5]. The V-ATPase a3 subunit is also known as TCIRG1 [28]. In previous reports, the deletion of TCIRG1 was shown to lead to an increase in bone density, which can lead to the development of osteosclerosis [29]. Although TCIRG1 is abnormally expressed in various malignancies, its expression in ccRCC and its potential prognostic significance remain unknown.

In this study, we first conducted a pan-cancer investigation and found that TCIRG1 was expressed significantly in several cancer types, and the results of GEO and UALCAN consistently showed that TCIRG1 expression was noticeably increased in KIRC samples. Furthermore, in KIRC survival analysis, high TCIRG1 expression was a poor predictive factor for OS, DSI, and PFI. To explore the role of TCIRG1 in the progression of ccRCC, we explored the relationship between the clinicopathological stage of the tumor and the prognostic value of TCIRG1 expression. TCIRG1 expression was found to be elevated in patients with advanced malignancies. TCIRG1 also showed excellent diagnostic efficacy for ccRCC, according to the ROC curves. Additionally, the findings from multivariate COX regression analysis and nomogram plots confirmed that elevated TCIRG1 expression is an essential indicator of an unfavorable OS in ccRCC. These results show that the TCIRG1 gene has prognostic value as a biomarker for KIRC.

To determine the molecular characteristics of TCIRG1, we identified mutations associated with the TCIRG1 expression levels in ccRCC. The findings demonstrated that *PBRM1* and *BAP1* mutations were different between the high- and low-TCIRG1 expression subgroups. Notably, the *PBRM1* mutant subgroup had lower TCIRG1 expression than the wild-type subgroup, whereas the *BAP1* mutant subgroup had higher TCIRG1 expression. According to findings from a previous study, ICIs improved survival in some patients with KIRC. After the whole-exome sequencing of metastatic KIRC in 35 patients, the clinical efficacy of ICI was found to be potentially associated with the loss of function and mutation in *PBRM1* [16]. Furthermore, the deletion of *BAP1*, a powerful tumor suppressor gene, imparted a significant adaptive advantage and lethal potential in KIRC clones [30]. To summarize, high TCIRG1 expression predicts fewer *PBRM1* mutations and more *BAP1* mutations, which may lead to poorer ICI responsiveness and a greater prevalence of KIRC.

To understand the function of TCIRG1 more comprehensively, we performed the enrichment analysis of co-expressed genes (FDR < 0.01, COR > 0.4) of TCIRG1. TCIRG1 expression was found to be associated with various immune-related pathways. The KEGG pathway enrichment analysis showed that TCIRG1 expression might be associated with various pathways, including the mTOR signaling pathway, *PDL1* expression, and the *PD1* checkpoint pathway, among others. Subsequently, we used the ESTIMATE algorithm to analyze the correlation between ImmuneScore, StromalScore, and estimated scores with TCIRG1 expression and survival from the TCGA-KIRC cohort [22]. The ImmuneScore and estimated scores were significantly higher in the high-TCIRG1 expression subgroup, and survival analysis revealed that survival was longer for high ImmuneScore and Estimatedscores than for low scores. Combined with these results, we demonstrate that the clinical outcome of patients with KIRC is significantly influenced by the immune microenvironment.

To further understand the immune function of TCIRG1, using CIBERSORT and the quanTiseq method, we investigated the relationship between TCIRG1 expression and TIL. We observed a high correlation between TCIRG1 expression and Tregs, CD8+ T cells, and NK cells. In addition, we observed that high TCIRG1 expression was negatively correlated with CD4 + T cells and neutrophil cells. These results suggest that TCIRG1 expression is associated with the level of immune cell infiltration. Tregs participate in tumor development and tumor immunity by inhibiting anti-tumor immunity [31]. In addition, evidence from previous studies indicates that CD4 + T cells are exhausted and may be critical in collective immune dysfunction [32]. Notably, in contrast to other tumor types, in RCC, increased CD8+ T cell infiltration is associated with poor renal cell prognosis and is more prevalent in metastatic renal cell carcinoma lesions. This mechanism requires further investigation [33].

Furthermore, we observed a strong correlation between TCIRG1 expression and immune marker gene expression based on data from the GEPIA and TIMER databases. Collectively, TCIRG1 expression can regulate immune cell infiltration, and TCIRG1 may interact within TME.

To explore the role of TCIRG1 in renal cancer immunotherapy, we first analyzed the potential for immune escape among different TCIRG1 expression subgroups and found that the high-TCIRG1 expression group was responsive to immunotherapy and had a higher propensity for immune escape, which also indicated that patients with low TCIRG1 expression might be more likely to benefit from immunotherapy. Subsequently, we also screened drugs suitable for patients in the high- and low-TCIRG1 expression subgroups based on the semi-inhibitory concentrations of the drugs. These results may have implications for guiding the clinical treatment of patients with KIRC.

Finally, the findings from our in vitro study showed that the knockdown of TCIRG1 significantly reduced the ability of renal cancer cells to migrate, which is consistent with the evidence from previous reports. However, the mechanism by which TCIRG1 promotes cancer cell migration and invasion has not been fully elucidated, and in a previous report, isoforms of the V-ATPase subunit have been shown to play a key role in cancer cell invasion. TCIRG1 is known to localize V-ATPase to the plasma membrane of specific acid-secreting cells. TCIRG1 is upregulated and critical in the invasion of melanoma cells, breast cancer cells, and hepatocellular carcinoma cells [9,10,11,12]. Overexpression of subunit a isoforms, particularly a3, may increase the trafficking of V-ATPase to the plasma membrane, providing tumor cells with an acidic tumor microenvironment and a slightly alkaline intracellular pH, which promotes cancer cell invasion. Overall, TCIRG1 could create a suitably low pH for histone proteases and matrix metalloproteinases (MMPs) to degrade the extracellular matrix and thus promote migration; TCIRG1 may also promote cancer cell migration through interaction and co-localization with the actin cytoskeleton [34], but the underlying mechanism remains unknown. In addition, V-ATPase function is required for the activation of the mechanistic target of rapamycin complex 1 (mTORC1) [35]. The mTOR pathway is one of the pathways that show consistent aberrant activation in ccRCC, and interestingly, we found that during enrichment, TCIRG1 showed a significant correlation, which may be another potential mechanism for predicting progression and poor prognosis in patients with ccRCC.

However, the present study has some limitations. First, our exploration of the role of TCIRG1 in KIRC was based only on pre-existing data available in databases such as TCGA and GEO. Therefore, a more comprehensive analysis as well as ex vivo experiments should be performed to elucidate the detailed molecular mechanisms underlying how TCIRG1 regulates the immunosuppressive TME. In addition, further insight is needed on the clinical significance of TCIRG1 in immunotherapy, which will also guide our future investigations.

In the present study, we demonstrated that the TCIRG1 gene is a promising immune-related prognostic biomarker for patients with KIRC.

## 5. Conclusions

High TCIRG1-expressing KIRC patients may incur a greater risk of tumor progression, and by the diversity of immune cell infiltration levels and status between the high- and low-TCIRG1 expression subgroups, patients with high TCIRG1 expression may receive a more precise immunotherapeutic strategy.

## Figures and Tables

**Figure 1 cancers-14-04583-f001:**
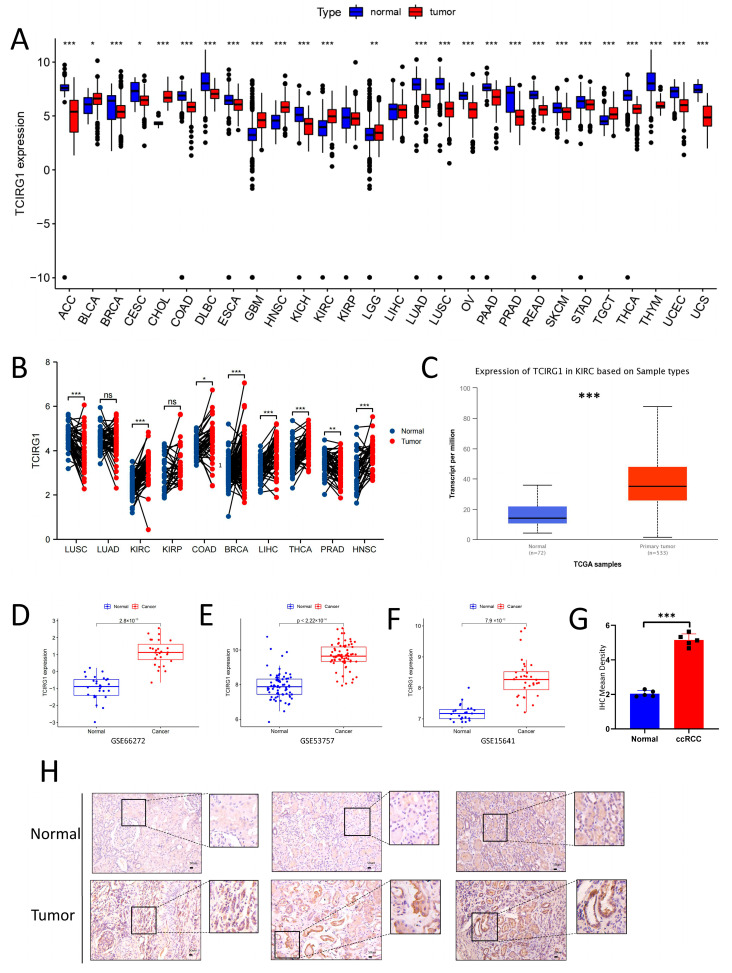
TCIRG1 expression in different tumors and their cancerous tissues. (**A**) TCIRG1 expression levels in TCGA tumor tissues and TCGA and GTEX normal tissues. (**B**) Comparison of TCIRG1 expression between TCGA tumor data and the corresponding TCGA normal data. (**C**) RNA expression levels of TCIRG1 in tumor and normal tissues in renal clear cell carcinoma (KIRC) samples from the UALCAN database. GEO database series, including TCIRG1 expression levels in GSE66272 (**D**) and GSE53757 (**E**), and GSE15641 (**F**–**H**) immunohistochemistry staining statistics results for normal kidney tissues and KIRC tissues. Scale bar: 50 μm. (* *p* < 0.05, ** *p* < 0.01, *** *p* < 0.001, ns indicates no statistical significance).

**Figure 2 cancers-14-04583-f002:**
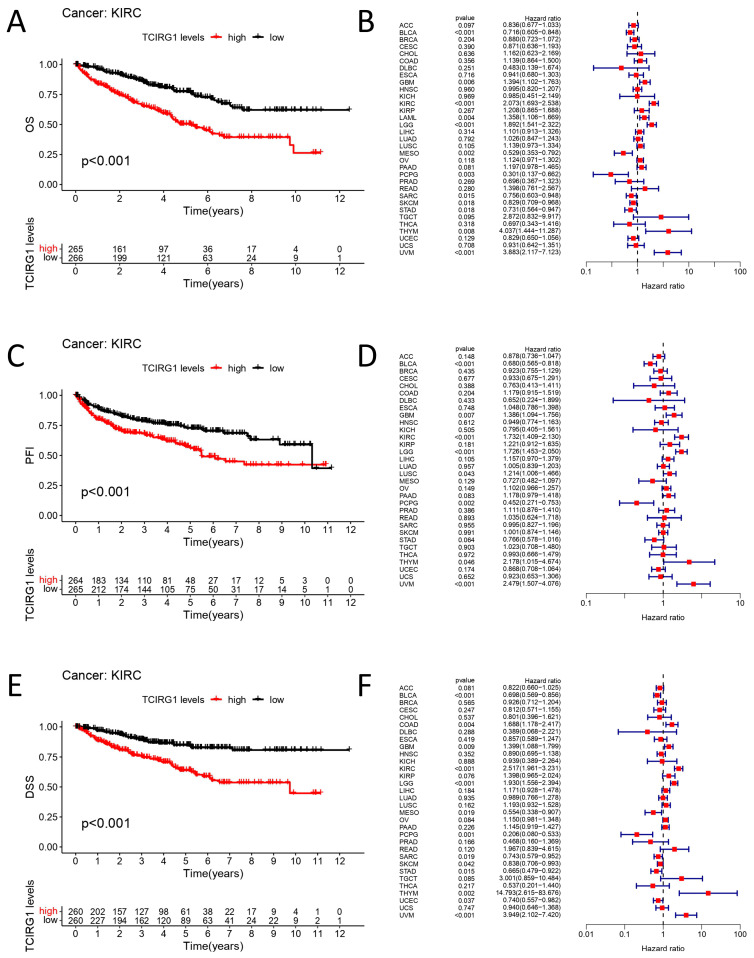
Correlation between the expression of TCIRG1 and tumor prognosis (**A**,**C**,**E**). Kaplan–Meier curves demonstrating OS, PFI, and DSS of patients in the high- and low-TCIRG1 expression groups in KIRC samples. Forest plots demonstrate the prognostic HR of TCIRG1 in different cancer subgroups in OS (**B**), PFI (**D**), and DSS (**F**).

**Figure 3 cancers-14-04583-f003:**
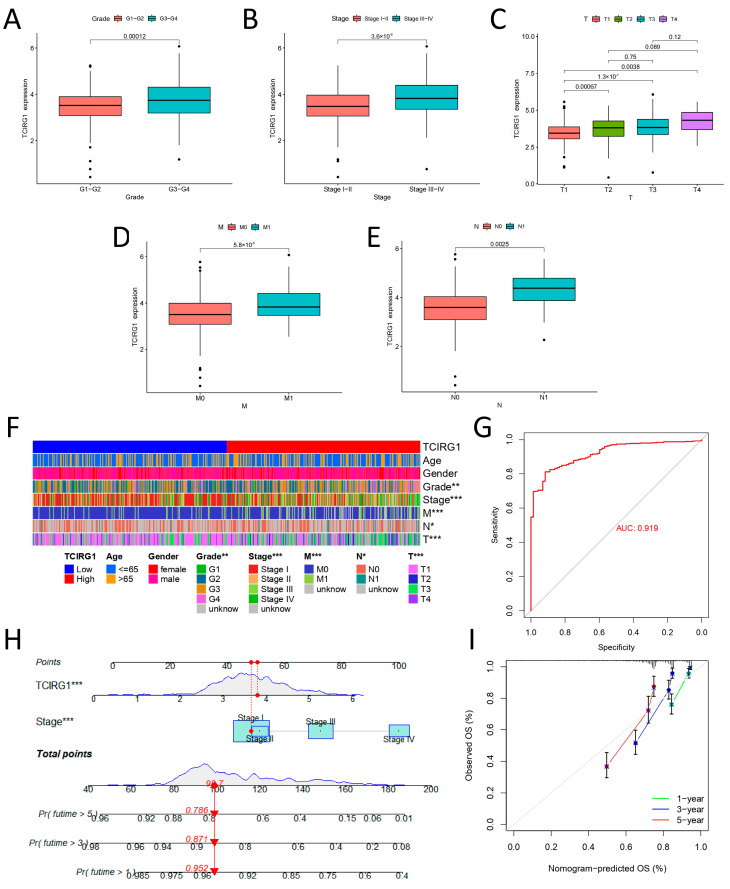
Correlation between high TCIRG1 expression and clinical traits. (**A**) Grade, (**B**) Stage, (**C**) T, (**D**) M, and (**E**) N correlation with TCIRG1 expression. (**F**) Heat map of correlation between clinical traits and TCIRG1 expression. (**G**) ROC curve to predict the diagnostic value of elevated TCIRG1. Nomogram for KIRC samples (**H**) and the calibration curve of the nomogram (**I**) for predicting OS at 1, 3, and 5 years. (* *p* < 0.05, ** *p* < 0.01, *** *p* < 0.001).

**Figure 4 cancers-14-04583-f004:**
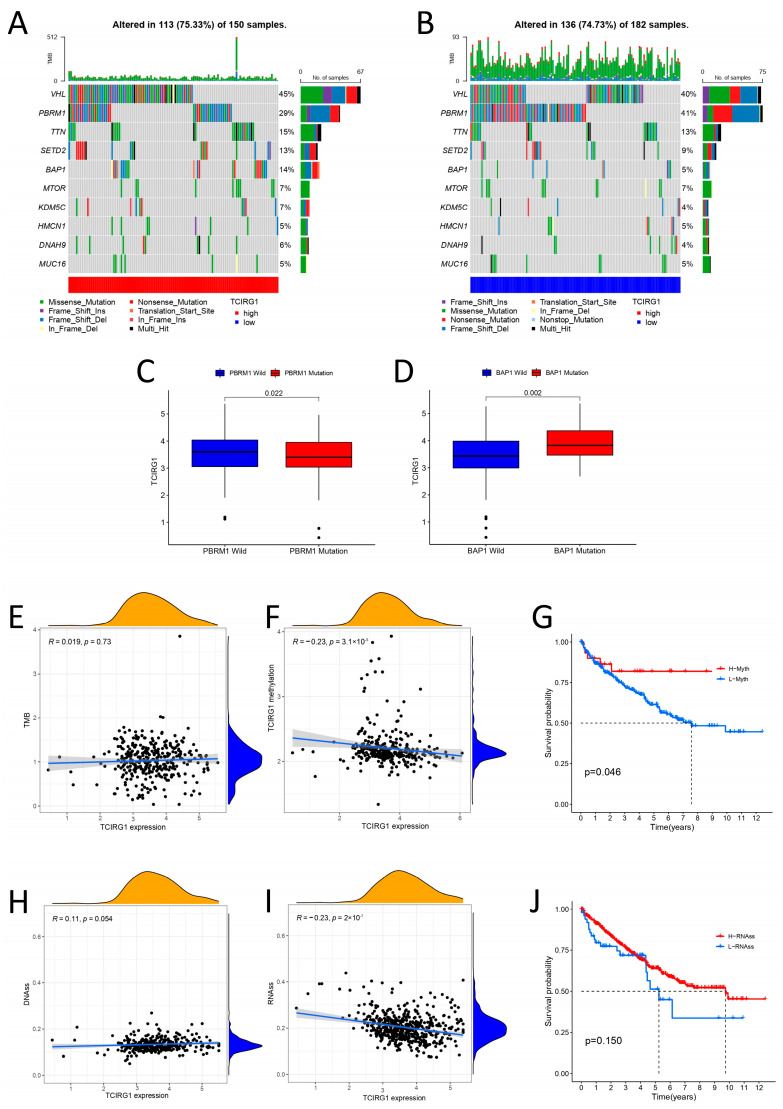
Molecular characterization of high- and low-TCIRG1 expression subgroups. (**A**,**B**) Distribution of the ten most commonly mutated genes in the high-TCIRG1 TCGA-KIRC subgroup. The top bar shows the TMB of each patient, and the right bar shows the different types of mutations. Differences in TCIRG1 expression between (**C**) wild-type and *PBRM1* mutant subgroups and (**D**) *BAP1* mutant subgroups. (**E**) Correlation analysis of TCIRG1 expression with TMB. (**F**) Relationship between TCIRG1 expression and DNA methylation. (**G**) Relationship between DNA methylation levels and OS. (**H,I**) Correlation analysis of TCIRG1 expression with cancer stemness DNAss and RNAss. (**J**) Correlation analysis of RNAss levels with OS.

**Figure 5 cancers-14-04583-f005:**
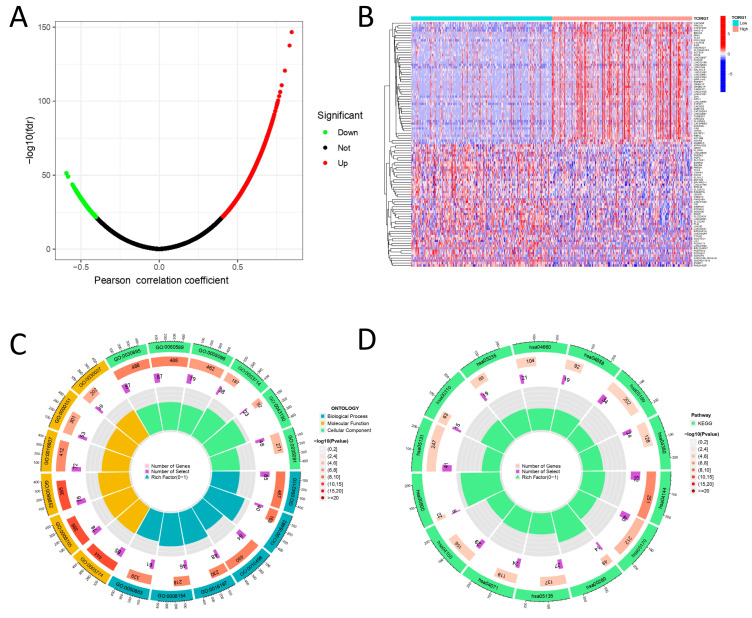
Enrichment analysis of co-expressed genes of TCIRG1 in TCGA-KIRC. (**A**) Volcano maps showing the co-expression of TCIRG1 genes (FDR < 0.01, COR > 0.4). (**B**) The top fifty genes with the strongest correlation are shown in the heat map. (**C**) GO pathway analysis of TCIRG1 co-expressed genes in KIRC. (**D**) KEGG pathway analysis of TCIRG1 co-expressed gene in KIRC.

**Figure 6 cancers-14-04583-f006:**
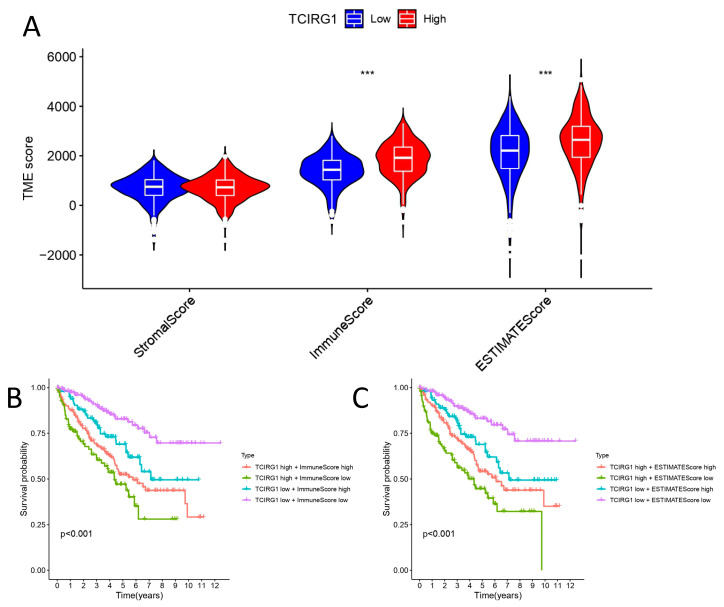
Correlation between TCIRG1 expression and tumor microenvironment score in KIRC. (**A**) BGN expression is associated with estimate score, Immunescore, and Stromalscore in renal cancer based on Pearson correlation analysis. (**B**,**C**) Relationship between OS of KIRC and Estimatedscore and Immunescore.(*** *p* < 0.001).

**Figure 7 cancers-14-04583-f007:**
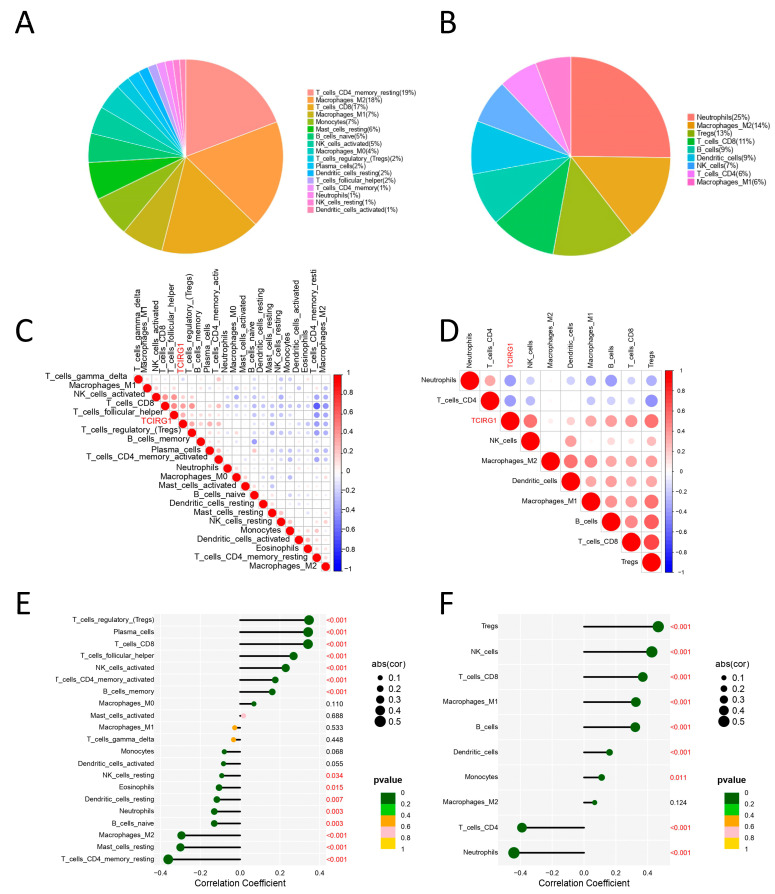
TIIC profile and correlation analysis in kidney cancer samples. Pie charts showing the estimated proportions of different types of TIICs in renal tumor samples predicted by (**A**) quanTIseq and (**B**) CIBERSOR. Pearson correlation matrix of the proportions of different TIICs in the microenvironment of renal cancer quantified by (**C**) quanTIseq and (**D**) CIBERSORT. The size of each bubble and the shading of each small colored box represents the corresponding correlation value between two cells. (**E**,**F**) Lollipop plots showing the correlation between TCIRG1 expression and different TIIC ratios analyzed by quanTIseq and CIBERSORT.

**Figure 8 cancers-14-04583-f008:**
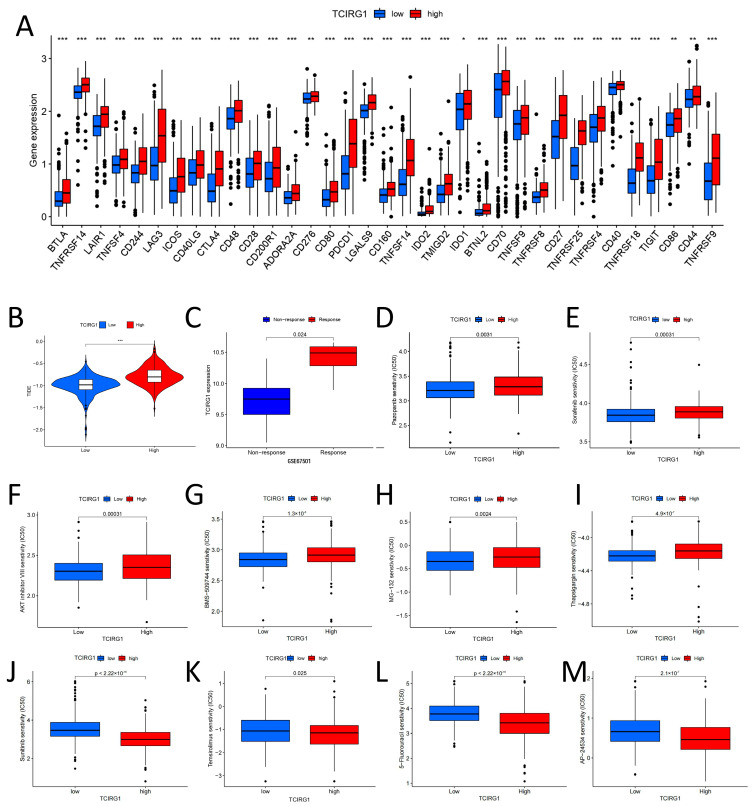
Division of subgroups according to TCIRG1 expression levels to predict potential immunotherapeutic responses in kidney cancer. (**A**) Immune checkpoint-associated genes are expressed in high- and low-TCIRG1 subpopulations. The Wilcoxon rank-sum test was used as a statistical significance test. (**B**) Tumor Immune Dysfunction and Exclusion (TIDE) score. (**C**) TCIRG1 expression differences between GSE67501-responding and non-responding groups. (**D**–**M**) IC50 differences between high- and low-TCIRG1 expression groups. (* *p* < 0.05, ** *p* < 0.01. *** *p* < 0.001).

**Figure 9 cancers-14-04583-f009:**
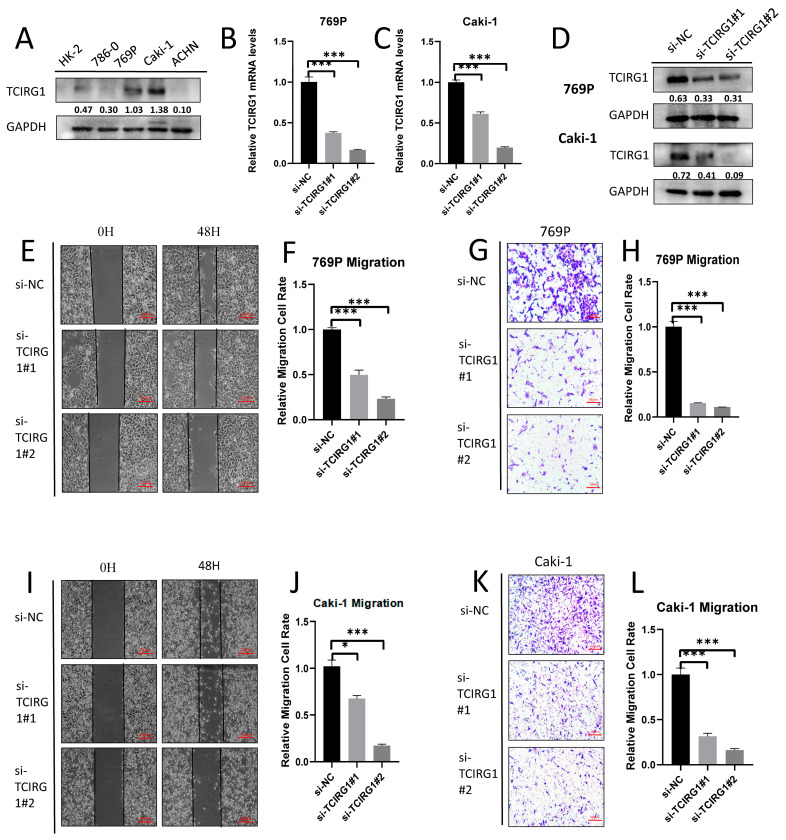
(**A**) Expression of TCIRG1 in different kidney cancer cell lines. TCIRG1 promotes the migratory ability of RCC cells. (**B**,**C**) qPCR Western Blot and (**D**) Western blotting showed that the expression of TCIRG1 was silenced by siRNA in 769P and caki1, respectively. Scratch wound-healing (**E**,**F**,**I**,**J**) and (**G**,**H**,**K**,**L**) transwell migration healing assays demonstrated TCIRG1-regulated migration ability. The scale bar of (E and I) is 250 μm, and the scale bar of (G and K) is 100 μm (* *p* < 0.05, *** *p* < 0.001).

**Table 1 cancers-14-04583-t001:** Analysis of OS using the Cox proportional hazards regression model.

Covariate	Univariate Analysis			Multivariate Analysis		
	HR	95% CI	*p*-Value	HR	95% CI	*p*-Value
Age (ref. ≤ 65 y)	1.029	1.016−1.043	<0.001	1.031	1.017–1.046	<0.001
Gender (ref. female)	0.943	0.689−1.291	0.716			
Grade (ref. G1–G4)	2.300	1.873−2.824	<0.001	1.417	1.125–1.786	0.003
Stage (ref.I–IV)	1.907	1.669−2.179	<0.001	1.635	1.400–1.909	<0.001
TCIRG1 (ref. low)	2.036	1.660−2.498	<0.001	1.546	1.259–1.899	<0.001

Significant differences as *p* < 0.001.

**Table 2 cancers-14-04583-t002:** TCIRG1 and immune cell biomarkers in RCC were correlated using GEPIA2 and Timer2.0.

		GEPIA2		Timer2.0	
Description	Gene Maker	Cor	*p*-Value	Cor	*p*-Value
CD8+ T cell	CD8A	0.37	<0.001	0.386	<0.001
	CD8B	0.37	<0.001	0.354	<0.001
T cell(genaral)	CD3D	0.43	<0.001	0.345	<0.001
	CD3E	0.43	<0.001	0.432	<0.001
	CD2	0.38	<0.001	0.439	<0.001
B cell	CD19	0.29	<0.001	0.399	<0.001
	CD79A	0.17	<0.001	0.406	<0.001
Monocyte	CD86	0.16	<0.001	0.231	<0.001
	CD115	0.25	<0.001	0.295	<0.001
TAM	CCL2	−0.044	0.31	0.016	<0.001
	CD68	−0.017	0.71	0.06	<0.001
	IL-10	0.045	0.31	0.141	<0.001
M1 Macrophage	INOS	0.01	0.82	−0.013	<0.001
	IRF5	0.3	<0.001	0.363	<0.001
	COX2	−0.065	0.14	−0.042	<0.001
M2 Macrophage	CD163	0.12	0.006	0.092	<0.001
	VSIG4	0.14	0.001	0.182	<0.001
	MS4A4A	0.084	0.055	0.129	<0.001
Neutroplis	CD66b	0.069	0.11	0.065	<0.001
	CD11b	0.093	0.034	0.282	<0.001
	CCR7	0.27	<0.001	0.289	<0.001
NK cell	KIR2DL1	0.034	0.44	0.07	<0.001
	KIR2DL3	0.071	0.1	0.102	<0.001
	KIR2DL4	0.21	<0.001	0.295	<0.001
	KIR3DL1	−0.024	0.59	0.029	<0.001
	KIR3DL2	0.15	<0.001	0.181	<0.001
	KIR3DL3	0.038	0.38	0.086	<0.001
	KIR2DS4	0.066	0.13	0.082	<0.001
Dentritic cell	HLA-DPB1	0.23	<0.001	0.272	<0.001
	HLA-DQB1	0.24	<0.001	0.241	<0.001
	HLA-DRA	0.16	<0.001	0.175	<0.001
	HLA-DPA1	0.13	<0.001	0.181	<0.001
	BCDA-1	0.038	0.38	0.09	<0.001
	BCDA-4	−0.14	<0.001	−0.128	<0.001
	CD11c	0.57	<0.001	0.508	<0.001
Th1	T-bet	0.42	<0.001	0.426	<0.001
	STAT1	0.22	<0.001	0.19	<0.001
	STAT4	0.45	<0.001	0.463	<0.001
	IFN-γ	0.37	<0.001	0.411	<0.001
	TNF-α	0.16	<0.001	0.227	<0.001
Th2	GATA3	0.067	0.12	0.32	<0.001
	STAT6	0.35	<0.001	0.163	<0.001
	STAT5A	0.33	<0.001	0.375	<0.001
	IL-13	0.23	<0.001	0.28	<0.001
Tfh	BCL6	0.18	<0.001	0.225	<0.001
	IL-21	0.22	<0.001	0.115	<0.001
Th17	STAT3	0.041	0.35	0.005	<0.001
	IL17A	0.024	0.59	0.046	<0.001
Treg	FOXP3	0.44	<0.001	0.468	<0.001
	CCR8	0.26	<0.001	0.295	<0.001
	STAT5B	−0.017	0.97	−0.074	<0.001
	TGFβ	0.24	<0.001	0.22	<0.001

*p*-values were calculated by using Pearson correlation analysis. Significant differences as *p* < 0.001.

## Data Availability

The original data covered in this study can be found in online databases, and detailed information is presented in the article/Appendix A.

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
