# Peer review of "Integrative Analysis Identifies TCIRG1 as a Potential Prognostic and Immunotherapy-Relevant Biomarker Associated with Malignant Cell Migration in Clear Cell Renal Cell Carcinoma"

_cancers, 2022, doi:10.3390/cancers14194583_

Round 1
Reviewer 1 Report
Xu et al., investigated the impact of TCIRG1 as a Potential Prognostic and Immunotherapy-relevant Biomarker. The study highlights the TCIRG1 was overexpressed in tumor tissue and predicted a significantly unfavorable clinical outcome.
What is the role of TCIRG1 in cancer and its relevant role in renal function?
The material and method section needs significant improvement. What do you mean by Materials and Metals? Authors introduced several abbreviations in this section, and introduced information on methods with no clear explanation in sections 2.1,2.3 and 2.4.
Poor presentation of of the results. Figure 9A and D are not clearly distinguishable, have the authors performed the densitometric analysis?
Discussion is a repetition of the results with no clear explanation of mechanism of action of TCIRG1.
Manuscript needs significant improvement of English.
Reviewer 2 Report
Thank authors for their job. Please consider some observations and suggestions for the sake to improve the paper:
I suggest to reflect in the abstract that the analysis was performed using TCGA and GEOR.
In Abstract, it is said: “High TCIRG1 may be associated with PBRM1 and BAP1 mutations…” (lines 36-37). In page 10, lines 285-286, “…higher TCIRG1 expression may result in fewer PBRM1 mutations and more BAP1 mutations (Figure 4C, D).” Please clarify and correct it in the Abstract.
Page 2, line 84: It should be: “Materials and Methods”.
At Discussion, page 18, lines 418-420, it is said: “Although anti-angiogenic agents have shown considerable efficacy in renal cell carcinoma, the role of ICIs remains limited in advanced renal cell carcinoma”. Although it is certain that not all patients benefit from immunotherapy, it is the same for anti-angiogenic drugs. In this way, it could appear that the latter (antiangiogenic) are more efficacious than the former (immunotherapy). Please consider to re-write the sentence.
Round 2
Reviewer 1 Report
I would like to thank you authors for clarifying the comments and incorporating the changes in the manuscript.